# Revision of the ARRIVE guidelines: rationale and scope

Nathalie Percie du Sert,[1] Viki Hurst,[1] Amrita Ahluwalia,[2] Sabina Alam,[3] Douglas G Altman,[4] Marc T Avey,[5] Monya Baker,[6] William Browne,[7] Alejandra Clark,[8] Innes C Cuthill,[7] Ulrich Dirnagl,[9] Michael Emerson,[10] Paul Garner,[11] David W Howells,[12] Natasha A Karp,[13] Catriona J MacCallum,[14] Malcolm Macleod,[15] Ole Petersen,[16] Frances Rawle,[17] Penny Reynolds,[18] Kieron Rooney,[19] Emily S Sena,[15] Shai D Silberberg,[20] Thomas Steckler,[21] Hanno Würbel,[22] Stephen T Holgate[23]

DGA is Deceased.

For numbered affiliations see end of article.

**Correspondence to**
Dr Nathalie Percie du Sert; nathalie.perciedusert@nc3rs.org.uk

## ABSTRACT

In 2010, the NC3Rs published the Animal Research: Reporting of In Vivo Experiments (ARRIVE) guidelines to improve the reporting of animal research. Despite considerable levels of support from the scientific community, the impact on the quality of reporting in animal research publications has been limited. This position paper highlights the strategy of an expert working group established to revise the guidelines and facilitate their uptake. The group's initial work will focus on three main areas: prioritisation of the ARRIVE items into a tiered system, development of an explanation and elaboration document, and revision of specific items.

Scientists, funders and the public are increasingly concerned about the reproducibility of preclinical research, including studies that use animals.[1] While the reasons for failing to reproduce the methods and findings of a study are complex and wide ranging, a lack of transparency stemming from poor reporting clearly contributes to the problem.[2] The NC3Rs coordinated the development of the Animal Research: Reporting of In Vivo Experiments (ARRIVE) reporting guidelines in 2010.[3] The guidelines consist of a 20-item checklist that covers the key information that should be described in a scientific publication. The goal is to ensure that the reader can assess the methodological rigour of the experiment, and other scientists can evaluate and reproduce the methods.

To date, over a thousand journals, funders and research institutes support and endorse ARRIVE.[4] The guidelines have contributed to the understanding of the issues that compromise reproducibility, and prompted actions to improve the situation. For example, major UK funders now explicitly mandate a comprehensive description of the study design, including plans to minimise experimental bias. Grant applicants must also explain how the number of animals to be used was decided and provide detailed statistical analysis plans to ensure that peer reviewers and panel members can fully assess the rigour and validity of the proposed research.[5]

Have the guidelines improved reporting? In the 7 years since the ARRIVE guidelines were published, researchers have sought to measure the impact of the guidelines on the quality of reporting,[6 7] with mixed results. A recent randomised controlled trial in *PLOS ONE*[8] showed that mandating the completion of an ARRIVE checklist with manuscript submission, with no additional emphasis on reporting during the editorial process, did not improve adherence to the guidelines in published papers. While these results are disappointing to those seeking immediate change, this study provides an evidence base to improve the guidelines, and ultimately the rigour and reproducibility of animal studies.

In the light of methodological advances in science and experience with the guidelines since their introduction, the NC3Rs convened an international working group to revise them. The authors here are members of the working group, and include funders, journal editors, statisticians, methodologists and animal researchers from academia and industry. The aim of this report is to provide readers and stakeholders with information about areas that we are currently working on to improve the ARRIVE guidelines during 2018. This work includes:

## PRIORITISING THE ITEMS OF THE ARRIVE GUIDELINES

Each of the 20 items of the ARRIVE guidelines are important for various reasons. For example, a description of study design (item

6), how the animals were allocated to groups (item 11) and how the sample size was chosen (item 10) are crucial to understand how reliable and robust the findings are. Similarly, items such as the experimental procedures (item 7) or animal characteristics (item 8) are important to ensure that papers contain enough information for others to replicate and build upon the study. Other ARRIVE items such as the scientific background (item 3) and relevance to other species (item 19) provide information about the context of the study.[3] In their current form, the guidelines do not lend themselves easily to retrospective evaluation; assessing whether a manuscript includes all 20 ARRIVE items necessitates operationalising the checklist into over a hundred separate elements.[9] To enable a more manageable approach for assessing the quality of reporting in manuscripts, we plan to organise the items in the ARRIVE guidelines into tiers reflecting different levels of priority; tier 1 items will include the most important items on which initial efforts from authors, reviewers and journals should focus. We will carry out a Delphi exercise,[10] to structure communications within the working group and with external stakeholders, and reach a consensus on the criteria defining the tiers, and on the most appropriate tier for each item. Importantly, the tiers will also enable a stepwise approach for journals and others to improve reporting standards, the objective being that ultimately all manuscripts will include all elements of the guidelines.

Prioritising subsets of items in this way will provide straightforward measures for journals, institutions and researchers. We anticipate that journals will continue to recommend that authors follow the ARRIVE guidelines in their entirety, to encourage comprehensive reporting. At the same time, focusing editorial efforts on a smaller number of key pieces of information that can be particularly scrutinised by editors and reviewers will enable a more rapid assessment of both individual manuscript quality, and the overall impact of their improvement strategies. This is an approach already used by some journals.[11 12] With the use of text mining and machine learning technologies, automating many of these checks is possible,[13] and coordinating work on top tier items will accelerate the development of tools to facilitate this.

## DEVELOPING AN EXPLANATION AND ELABORATION DOCUMENT

Understanding the rationale for a set of guidelines is essential for securing support from the scientific community. The Consolidated Standards of Reporting Trials (CONSORT) statement, for example, has been accompanied by an Explanation and Elaboration document since its second iteration in 2001.[14] This summarises the evidence behind each item of the guidelines and explains why each item is important to include in a manuscript.

A recent survey of in vivo researchers carried out by the NC3Rs[15] highlighted that the main reason for authors not including an ARRIVE item in a manuscript was because they did not think it was necessary to disclose that information. Incomplete reporting is also exacerbated by the fact that some of the concepts included in the guidelines, such as measures to reduce bias, are not well understood by researchers or not considered relevant for their own research.[16] To address this, we are now developing an Explanation and Elaboration document for the ARRIVE guidelines. This document will provide explanations and definitions for technical terms, empirical evidence in support of each ARRIVE item, as well as examples from the published literature on how authors might report items. Following publication of the document, the information will be made readily accessible via a dedicated ARRIVE website.

## REVISING THE GUIDELINES

We are reviewing specific ARRIVE items to ensure that the guidance provided is in line with the current best evidence. Where evidence is lacking we will seek to develop it. The revision is an opportunity to improve the clarity of individual items, ensure their relevance across the breadth of in vivo research and enhance the logical flow of information within the guidelines. In recent years, scientific organisations such as publishers and funders have produced additional guidance to improve the reporting of preclinical research (eg, NINDS's call for greater transparency in preclinical research,[17] NIH's Principles and Guidelines for Reporting Preclinical Research,[18] *Nature*'s Reporting Life Sciences Research checklist,[19] *Cell* guidelines,[20] *British Journal of Pharmacology*'s guidance for reporting experimental design and analysis[21]). Such guidelines will be taken into consideration in the revision. The ARRIVE guidelines are not intended to supersede journal or model-specific guidelines but the level of support from funders and journals puts ARRIVE in a unique position to serve as the basis for more specialised guidelines. The publishing landscape has also evolved over the last decade and the revised guidelines will reflect these changes by providing advice on emerging best practice. Additionally, external stakeholders with expertise in preclinical research reporting will be able to suggest further revisions via the above-mentioned Delphi exercise.

The scope of the ARRIVE guidelines is broad; they are designed to be flexible and accommodate the reporting of comparative studies in a wide range of research areas. As such, the existing guidelines were formulated to provide general advice for heterogeneous study types. However, recent calls have been made to encourage researchers to explicitly distinguish between exploratory and hypothesis-testing studies.[22 23] For hypothesis-testing studies that are using inferential statistics, the manuscript would be expected to describe the primary and secondary outcome measures, the parameters used in the sample size calculation, whether the study protocol was preregistered, and if so, where it can be found. Exploratory studies, on the other hand, are designed to generate hypotheses; they might confer the same importance to all outcomes measured, might justify the sample size based on feasibility or experience and would report only descriptive statistics. Thus, the reporting requirements for exploratory and hypothesis-testing studies

can differ, and the revision will ensure that the updated guidelines provide adequate advice for both.

## FINAL REMARKS

Revising the guidelines is just the first step; their primary goal is to improve transparency and the standards of reporting, but transparent reporting can be used to address common weaknesses in the design and conduct of animal research and encourage researchers to adopt more rigorous scientific practices. Ultimately, the ARRIVE guidelines will form the basis for a powerful suite of tools and resources that provide optimal support for researchers to improve the design, conduct and reporting of in vivo research; this will also benefit research users and stakeholders tasked with assessing the quality and translational value of preclinical research.

Improving reporting should be a community-wide effort, and the working group recognises the importance of engaging others in the evolution of the ARRIVE guidelines. It is essential that this endeavour includes scientists from a range of research fields and countries, and we encourage the community to share their experience and views.

**Author affiliations**
[1]NC3Rs, London, UK
[2]Queen Mary University of London, London, UK
[3]F1000 Research, London, UK
[4]University of Oxford, Oxford, UK
[5]ICF Canada, Ottawa, Ontario, Canada
[6]Nature, San Francisco, California, USA
[7]University of Bristol, Bristol, UK
[8]PLOS ONE, London, UK
[9]QUEST–Center for Transforming Biomedical Research, Berlin Institute of Health (BIH), Berlin, Germany
[10]Imperial College London, London, UK
[11]Liverpool School of Tropical Medicine, Liverpool, UK
[12]University of Tasmania, Hobart, Tasmania, Australia
[13]Quantitative Biology, Discovery Science, IMED Biotech Unit, Cambridge, UK
[14]Hindawi, London, UK
[15]Centre for Clinical Brain Sciences, University of Edinburgh, Edinburgh, UK
[16]Cardiff University, Cardiff, UK
[17]Medical Research Council, London, UK
[18]University of Florida, Gainesville, Florida, USA
[19]University of Sydney, Sydney, New South Wales, Australia
[20]National Institute of Neurological Disorders and Stroke, Bethesda, Maryland, USA
[21]Janssen Pharmaceutica NV, Beerse, Belgium
[22]University of Bern, Bern, Switzerland
[23]University of Southampton, Southampton, UK

**Acknowledgements** We would like to acknowledge the late Doug Altman's instrumental contribution to this project. Doug collaborated with the NC3Rs on the work leading up to the development of the ARRIVE guidelines, he was an author on the original guidelines and an active member of the present working group to revise them. It was a pleasure to work with him over the years and he will be sorely missed.

**Contributors** NPS: Conceptualization(Equal) Project administration(Supporting) Writing – original draft(Lead) Writing – review & editing(Equal). VH: Conceptualization(Equal) Project administration(Lead) Writing – review & editing(Equal). AA: Conceptualization(Equal) Writing – review & editing(Equal). SA: Conceptualization(Equal) Writing – review & editing(Equal). DA: Conceptualization(Equal) Writing – review & editing(Equal). MA:Conceptualization(Equal) Writing – review & editing(Equal). MB: Conceptualization(Equal) Writing – review & editing(Equal). WB: Conceptualization(Equal) Writing – review &
editing(Equal). AC: Conceptualization(Equal) Writing – review & editing(Equal). IC: Conceptualization(Equal) Writing – review & editing(Equal). UD: Conceptualization(Equal) Writing – review & editing(Equal). ME: Conceptualization(Equal) Writing – review & editing(Equal). PG: Conceptualization(Equal) Writing – review & editing(Equal). DH: Conceptualization(Equal) Writing – review & editing(Equal). NK: Conceptualization(Equal) Writing – review & editing(Equal). MM: Conceptualization(Equal) Writing – review & editing(Equal). OP: Conceptualization(Equal) Writing – review & editing(Equal). FR:Conceptualization(Equal) Writing – review & editing(Equal). PR: Conceptualization(Equal) Writing – review & editing(Equal). KR: Conceptualization(Equal) Writing – review & editing(Equal). ES: Conceptualization(Equal) Writing – review & editing(Equal). ES: Conceptualization(Equal) Writing – review & editing(Equal). SS:Conceptualization(Equal) Writing – review & editing(Equal). TS: Conceptualization(Equal) Writing – review & editing(Equal). HW: Conceptualization(Equal) Writing – review & editing(Equal). SH: Conceptualization(Equal) Writing – review & editing(Equal).

**Funding** The authors have not declared a specific grant for this research from any funding agency in the public, commercial or not-for-profit sectors.

**Competing interests** AA: editor in chief of the British Journal of Pharmacology. DGA, WB, ICC and ME: authors of the original ARRIVE guidelines. WB: professor of Statistics at the University of Bristol, consults for HEFCE and the UK Home Office, work funded by many of the UK research councils, Defra, the British Academy and the RSPCA, serves on the Independent Statistical Standing Committee of the funder CHDI foundation, chair of the members committee of the Cathedral Schools Trust, a multiple academy trust and a governor of Wrington Church of England primary school. AC, CJM, MMcL and ESS: involved in the IICARus trial. ME, MMcL and ESS: have received funding from NC3Rs. ME: sits on the MRC ERPIC panel. STH: chair of the NC3Rs board, trusteeship of the BLF, Kennedy Trust, DSRU and CRUK, member of Governing Board, Nuffield Council of Bioethics, member Science Panel for Health (EU H2020), founder and NEB Director Synairgen, consultant Novartis, Teva and AZ, chair MRC/GSK EMINENT Collaboration. VH and NPdS: NC3Rs staff, role includes promoting the ARRIVE guidelines. CJMcC: shareholdings in Hindawi, on the publishing board of the Royal Society, on the EU Open Science policy platform. MMcL, NPdS, CJMcC, ESS, TS and HW: members of EQIPD. MMcL: member of the Animals in Science Committee. NPdS and TS: associate editors of BMJ Open Science. OP: vice president of Academia Europaea, senior executive editor of the Journal of Physiology, member of the Board of the European Commission's SAPEA (Science Advice for Policy by European Academies). FR: NC3Rs board member, shareholdings in AstraZeneca and GSK. ESS: editor in chief of BMJ Open Science. SDS: role is to provide expertise and does not represent the opinion of the NIH. TS: shareholdings in Johnson & Johnson. SA, MTA, MB, UD, PG, DWH, NAK, PR and KR declared no conflict of interest. ESS is editor in chief of this journal and was not part of the decision-making process for this manuscript.

**Provenance and peer review** Not commissioned, externally peer reviewed.

**Data sharing statement** There are no data in this work.

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
