## [Reviewer comments · BMJ Open Science]

ARTICLE DETAILS

TITLE (PROVISIONAL)	Revision of the ARRIVE guidelines: Rationale and scope
AUTHORS	Nathalie Percie du Sert (Corresponding Author) Viki Hurst Amrita Ahluwalia Sabina Alam Douglas G Altman Marc Thomas Avey Monya Baker William John Browne Alejandra Monica Clark Innes Cuthill Ulrich Dirnagl Michael Emerson Paul Adrian Garner David Howells Natasha A Karp Catriona MacCallum Malcolm Macleod Ole Petersen Frances Rawle Penny S Reynolds Kieron Rooney Emily Sena Shai David Silberberg Thomas Steckler Hanno Wuerbel Stephen T Holgate

VERSION 1 - REVIEW

REVIEWER 1	Mhairi Macrae Glasgow University Conflict of interest: I have collaborated in the past with a number of the authors: as a member of Multi-PART and my involvement with NC3Rs on producing the IMPROVE guidelines for in vivo research on stroke.
REVIEW RETURNED	22/02/18

GENERAL COMMENTS	A revision of the ARRIVE guidelines is both logical and timely given the 8 years since the guidelines were first published and recent analyses of the literature which provides evidence for a lower than expected improvement in the reporting of in vivo research. My own view is that the ARRIVE checklist is easy to understand, with some items more straightforward to address in a manuscript than others. Consequently manuscripts are submitted with some but rarely all 20 items included. Some of the
--

	responsibility for the limited success of the guidelines must therefore be borne by the editors and publishers who have not put processes in place to check that the guidelines have been followed. This manuscript clearly lays out issues which may have led to the limited success of ARRIVE, and a plan for the restructuring and refinement of the guidelines based on feedback from the NC3Rs survey and reports in the literature. The plans for a tiered system and more details and justification for the individual checklist items should help authors comply more fully with the guidelines when writing their manuscripts. However, the inclusion of journal editors, publishers and representatives of funding bodies within the expert working group will be essential for the revised guidelines to have the greatest impact as they have the greatest power and duty to ensure that the guidelines are adhered to in funding applications and published articles. New machine learning technologies designed for automated checking of manuscripts must be fit for purpose for the expected improvements in reporting unless publishers are willing to employ extra staff to undertake the task of checking manuscripts. Whether machine learning technologies or specific staff are employed, there will be an additional cost to improve the quality of the published literature so early dialogue and lobbying of the main publishers would be advisable to have these systems in place to optimise the success of the revised guidelines. The revised guidelines should be targeted at both journal editors and publishers as well as manuscript authors and should provide as much information as possible on automated checking of manuscripts in order to maximise the use of this technology in supporting the revised guidelines. I believe journal editors and publishers are the people capable of making the greatest improvement in adherence to the guidelines in the future.
--	--

REVIEWER 2	Adrian Smith Norecopa Conflict of Interest: I would like to indicate one "relationship" to the ARRIVE guidelines which you should be aware of, so you can decide whether it constitutes a conflict of interest or not. I am lead author of the PREPARE guidelines for planning animal experiments: https://norecopa.no/PREPARE PREPARE addresses animal experiments from the other end (planning) while ARRIVE addresses (mainly) reporting. However, in the Speakers Notes accompanying a presentation about ARRIVE on the NC3Rs website, the NC3Rs make the claim that ARRIVE 'provides a logical checklist with all the things that need to be considered when designing an animal experiment' (https://www.nc3rs.org.uk/sites/default/files/documents/
---

	Guidelines/ARRIVE%20Guidelines%20Speaker%20Notes.pdf). We do not agree with this statement (that ARRIVE covers everything), which is why we published the PREPARE guidelines.
REVIEW RETURNED	11/04/18

GENERAL COMMENTS	The authors are to be commended for embarking on this task. In particular I welcome their final remarks concerning their intention to engage others in the evolution of the ARRIVE guidelines.
--

VERSION 1 – AUTHOR RESPONSE

Thank you for reviewing our paper and considering it for publication. We thank the reviewers for their supportive comments. We have modified the manuscript, in line with the suggestions and believe we have addressed all the queries.

We have clarified how other guidelines and ongoing activities will feed into the revision on the ARRIVE guidelines. We also specified how stakeholders who are not currently part of the working group will be able to contribute to the revision. The following text was added to the section on revising the guidelines:

“In recent years, scientific organisations such as publishers and funders have produced additional guidance to improve the reporting of preclinical research (e.g. NINDS’s call for greater transparency in pre-clinical research¹⁷, NIH’s Principles and Guidelines for Reporting Preclinical Research¹⁸, Nature’s Reporting Life Sciences Research checklist¹⁹, Cell guidelines²⁰, British Journal of Pharmacology’s guidance for reporting experimental design and analysis²¹). Such guidelines will be taken into consideration in the revision. The ARRIVE guidelines are not intended to supersede journal or model-specific guidelines but the level of support from funders and journals puts ARRIVE in a unique position to serve as the basis for more specialised guidelines. The publishing landscape has also evolved over the last decade and the revised guidelines will reflect these changes by providing advice on emerging best practice. Additionally, external stakeholders with expertise in preclinical research reporting will be able to suggest further revisions via the above-mentioned Delphi exercise.”